# Sonic Hedgehog Regulates Bone Fracture Healing

**DOI:** 10.3390/ijms21020677

**Published:** 2020-01-20

**Authors:** Hiroaki Takebe, Nazmus Shalehin, Akihiro Hosoya, Tsuyoshi Shimo, Kazuharu Irie

**Affiliations:** 1Division of Histology, Department of Oral Growth and Development, School of Dentistry, Health Sciences University of Hokkaido, 1757 Kanazawa, Ishikari-Tobetsu Hokkaido 061-0293, Japan; shalehin@hoku-iryo-u.ac.jp (N.S.); hosoya@hoku-iryo-u.ac.jp (A.H.); irie@hoku-iryo-u.ac.jp (K.I.); 2Division of Reconstructive Surgery for Oral and Maxillofacial Region, Department of Human Biology and Pathophysiology, School of Dentistry, Health Sciences University of Hokkaido, 1757 Kanazawa, Ishikari-Tobetsu Hokkaido 061-0293, Japan; shimotsu@hoku-iryo-u.ac.jp

**Keywords:** sonic hedgehog, stem cell, animal experiment, fracture healing

## Abstract

Bone fracture healing involves the combination of intramembranous and endochondral ossification. It is known that Indian hedgehog (Ihh) promotes chondrogenesis during fracture healing. Meanwhile, Sonic hedgehog (Shh), which is involved in ontogeny, has been reported to be involved in fracture healing, but the details had not been clarified. In this study, we demonstrated that Shh participated in fracture healing. Six-week-old Sprague–Dawley rats and Gli-CreER^T2^; tdTomato mice were used in this study. The right rib bones of experimental animals were fractured. The localization of Shh and Gli1 during fracture healing was examined. The localization of Gli1 progeny cells and osterix (Osx)-positive cells was similar during fracture healing. Runt-related transcription factor 2 (Runx2) and Osx, both of which are osteoblast markers, were observed on the surface of the new bone matrix and chondrocytes on day seven after fracture. Shh and Gli1 were co-localized with Runx2 and Osx. These findings suggest that Shh is involved in intramembranous and endochondral ossification during fracture healing.

## 1. Introduction

The fracture healing process consists of four overlapping phases, namely, inflammation, proliferation, callus formation, and bone remodeling. Immediately following fracture, the injury initiates an inflammatory response that is necessary to promote healing. The response induces the development of a hematoma, which consists of cells from both peripheral blood vessels and bone marrow. The hematoma coagulates between and around the fracture site and within the bone marrow, providing a template for callus formation [1]. Vascularization supplies mesenchymal stem cells (MSCs), which differentiate into chondrocytes or osteoblasts simultaneously with cartilage tissue development (proliferation phase) [2,3]. The cartilage matrix begins to form at the fractured bone gap during the callus formation phase. Meanwhile, intramembranous ossification occurs internal to the periosteum adjacent to the fracture line and forms the bone matrix [4]. MSCs directly differentiate into osteoblasts at the fracture site along the proximal and distal edges of fractured bone during intramembranous ossification. After cartilage tissue maturation, new bone formation is initiated as the cartilage tissue is resorbed and vascularization is induced to replace the cartilage tissue with bone. It has also been reported that primary bone formation is initiated peripheral to the newly formed cartilage region at the fractured bone site [5]. The bone remodeling phase recapitulates embryonic bone development with a combination of cellular proliferation and differentiation, increasing the cellular volume and matrix deposition [1]. Finally, remodeling of the hard callus into a lamellar bone structure occurs (bone remodeling phase).

The biological process occurring during bone fracture healing is regulated by several signaling molecules. Hedgehog (HH) proteins are among the signaling molecules required for endochondral bone formation during embryonic development, and they regulate bone homeostasis by controlling MSC proliferation [6,7]. HH signaling is also involved in the regulation of MSC proliferation in adult tissues. Aberrant activation of HH pathways has been linked to multiple types of human cancer [7]. These pathways are also activated during intramembranous and endochondral ossification in the fracture healing process, but it is not clear if they are involved in the healing process [5]. HH signaling pathways play critical roles in developmental processes and in the postnatal homeostasis of many tissues, including bone and cartilage. The HH family of intercellular signaling proteins plays important roles in regulating the development of many tissues and organs. Their name is derived from the observation of a hedgehog-like appearance in *Drosophila* embryos with genetic mutations that block their action. Three types of HH proteins have been reported in mammals, namely Sonic HH (Shh), Indian HH (Ihh), and Desert HH (Dhh). Ihh is up-regulated during the initial stage of fracture repair, and it regulates differentiation indirectly by controlling cartilage development at the fracture site. Ihh regulates osteoblast differentiation indirectly by controlling cartilage development [8]. In general, Shh acts in the early stages of development to regulate patterning and growth [9]. Recently, several studies reported that Shh might be related to fracture healing [10,11]. Following the inactivation of HH signaling, the activity of Smo is inhibited by a receptor known as Patched (Ptch). Binding of the HH ligand Ptch relieves the inhibition of Smo, and activated Smo blocks the proteolysis of Gli proteins in the cytoplasm and promotes their dissociation from suppressor of fused (SuFu). Following dissociation from SuFu, activated Gli proteins translocate into the nucleus and promote the expression of Hh target genes, including *Gli1* [9,12]. Gli1 positivity has been identified as a marker for MSCs [13]. Another study uncovered that Gli1 is involved in osteoblast differentiation [14]. However, it is unclear that whether Shh proteins are involved in fracture healing. In this study, we demonstrated that Shh protein and the related proteins Smo and Gli1 were involved in osteoblast differentiation at the fracture healing site via immunohistochemical analysis.

## 2. Results and Discussion

In this study, we hypothesized that Shh is related to the healing process of fractures and investigated and compared the positive localization of Runx2 and Osx, which appear during the fracture repair process, with that of Shh and its downstream factor Gli1. Runt-related transcription factor 2 (Runx2), which is an essential factor for bone formation, is expressed very early in skeletal development. Osterix (Osx) is activated downstream of Runx2 during osteoblastic lineage differentiation [15,16]. On the day of fracture (day 0), a few Runx2-positive and Osx-positive cells were observed on the bone surface in the periosteum (Figure 1a,c). Shh-positive cells were rarely observed in the periosteum on day 0 (Figure 1b). Furthermore, Gli1-positive cells were also rarely observed (Figure 1d). However, Shh and Gli1 positivity were localized to osteocytes in the bone matrix. These results indicated that Shh signaling occurred in osteocytes but not in undifferentiated cells in the periosteum. Moreover, in this study, we traced the fate of Gli1-positive cells in Gli1-Cre recombinase-mutated estrogen receptors (CreER^T2^); tdTomato mice on day seven after fracture by administering tamoxifen. Previous reports demonstrated that 3 days are required for Cre activation after tamoxifen administration [8]. In our genetically modified mouse system, both Gli1-positive cells and their progeny were permanently marked by red fluorescent protein expression. Gli1-CreER^T2^; tdTomato mice, which are transgenic for the *Gli1-CreER^T2^/Rosa26-loxP-stop-loxP-tdTomato* gene, were used to generate Gli1-positive and progeny cells through lineage-tracing analysis. Gli1-positive cells expressed the CreER^T2^. CreER^T2^ has the function of specifically recognizing and removing the LoxP site. Moreover, CreER^T2^ binds to tamoxifen but not to natural estrogens. Gli1-positive cells were found to express tomato red fluorescence after tamoxifen administration. Since tomato fluorescence is expressed permanently, not only Gli1-positive cells but also progeny cells were found to express tomato red fluorescence [17].

New cartilage matrix formed around the fracture site (Figure 1e). The localization of Gli1 progeny cells and Osx-positive cells was examined at the fracture healing site (Figure 1e*). Gli1 positivity was observed in chondrocytes and the perichondrium around the cartilage matrix at the fracture site (Figure 1e*). Osx positivity was localized in cells surrounding new cartilage matrix. The merged image of Gli1-positive and Osx-positive areas indicated that most Osx-positive cells were co-localized with Gli1-positive cells (Figure 1e*). These results indicate that Gli-positive cells and their progeny cells might differentiate into osteoblasts after bone fracture. This result was consistent with another report that Gli1 marked a major skeletal progenitor pool contributing to both bone and cartilage formation during bone fracture healing in postnatal mice [8]. 

On day 1, hematoma and granulation tissue were observed at the bone fracture gap. Runx2-positive and Osx-positive cells were observed extensively in the remaining periosteum near the fracture site (Figure 2a*,b*). However, few Runx2-positive and Osx-positive cells were noted in the intact periosteum far from the fracture site (Figure 2a**,b**). Runx2-positive and Osx-positive cell numbers in the periosteum near the fracture site were remarkably higher than those far from the fracture site (Figure 3A,B). These results indicated that MSCs committed to osteoblast or chondroblast differentiation participated in intramembranous and endochondral ossification only near the fracture site. 

Large numbers of Shh-positive and Gli1-positive cells were also found in the periosteum only near the fracture site (Figure 4a*,b*). Their numbers increased on day one in the periosteum adjacent to the fracture site compared with the number of Shh-positive and Gli1-positive cells on day 0 (Figure 5A,B). Shh-positive and Gli1-positive cell numbers in the periosteum near the fracture site were remarkably higher than those far from the fracture site (Figure 5A,B). These results indicate that Shh and Gli1, which emerge after bone fracture in the periosteum, might be associated with osteoblast differentiation. 

On day 7, a newly formed cartilage matrix was observed in the fracture gap (Figure 6a). It has been reported that endochondral ossification is observed in the fracture gap at that time [18,19]. Sox9 positivity was observed in chondrocytes as well as on the bone matrix surface (Figure 6b*). Runx2-positive and Osx-positive cells were observed on the surfaces of newly formed bone matrix and chondrocytes (Figure 6b**,c*,c**). Shh-positive and Gli1-positive cells were also observed on the surfaces of newly formed bone matrix and chondrocytes (Figure 6d*,d**,e*,e**). In addition, new bone matrix extending from the proximal and distal edges of the fractured bone surface, which is termed intramembranous ossification, was observed (Figure 6a). A large number of Osx-positive cells were also observed on the surface of newly formed bone extending from the proximal and distal edges. Osx-positive cells were localized on the new bone surface around the newly formed cartilage matrix (Figure 6c*). This result is consistent with another report in which MSCs directly differentiated into osteoblasts in the perichondrium around the cartilage matrix after bone fracture, resulting in bone formation [20]. Shh-positive and Gli1-positive cells localized along the surface of newly formed bone (Figure 6d*,d**,e*,e**). These results indicate that the Shh–Gli1 signaling pathway might regulate intramembranous and endochondral ossification at the fracture site. 

On day 14, newly formed cartilage matrix at the fracture site began to resorb, and it was replaced by newly formed bone known as primary bone (Figure 7a) [21]. In the resorbed cartilage area, many Osx-positive and cathepsin K (CathK)-positive cells were observed (Figure 7b,c). Positivity for osteopontin (OPN), which is a bone matrix component, was observed around the resorbed cartilage matrix (Figure 7d) [22]. Shh and Gli1 positivity localized around the resorbed cartilage matrix (Figure 7e,f). These results indicate that the Shh–Gli1 signaling pathway participates in new bone formation by osteoblasts. 

## 3. Materials and Methods

### 3.1. Experimental Animals

Twenty-six-week-old male Sprague–Dawley rats (Hokudo, Sapporo, Japan) and three Gli-CreER^T2^; tdTomato male mice (Jackson Laboratory, Bar Harbor, ME, USA) were used. All experimental animals were maintained in a specific pathogen-free facility. All experiments were approved and performed according to guidelines set forth by the Animal Ethics Committee of the Health Sciences University of Hokkaido (The ethical permission code and permission date: 19-028, 8 March, 2019 and 19-045, 29 March 2019).

### 3.2. Tamoxifen Administration

Gli1-CreER^T2^; tdTomato mice were injected intraperitoneally with tamoxifen (Sigma-Aldrich, St. Louis, MO, USA) once daily for 3 consecutive days (40 mg/mL, dissolved in corn oil). 

### 3.3. Fracture Experiment

The right eighth rib of each experimental animal was fractured as previously described [10]. Briefly, each experimental animal was anesthetized, and the eighth rib on the right side was exposed and cut vertical to the axis with scissors. As a control, the right eighth rib of select animals was similarly exposed but not fractured. 

### 3.4. Tissue Preparation

The animals were anesthetized subcutaneously with pentobarbital sodium (40 mg/kg) and killed via cervical dislocation. The ribs of Gli1-CreER^T2^; tdTomato mice were collected 7 days after fracture and immediately frozen at −80 °C. Each sample was embedded in 5 % carboxymethyl cellulose (CMC) gel (Section-Lab Co. Ltd., Tokyo, Japan). Each frozen CMC sample was covered with polyvinylidene chloride film (Section-Lab Co. Ltd.) and sagittally sectioned at a thickness of 5 µm. The ribs of rats were collected 0, 1, 7, and 14 days after fracture and fixed in 4.0% paraformaldehyde in 0.1 M phosphate buffer (pH 7.4) overnight at 4 °C. Specimens were demineralized via immersion in 10% ethylenediaminetetraacetic acid (pH 7.4) for 4 weeks at 4 °C. Following demineralization, the specimens were embedded in paraffin and sectioned at a thickness of 5 µm. 

### 3.5. Immunohistochemistry

For immunohistochemistry, the dehydrated sections were treated with 0.3% H_2_O_2_ in phosphate-buffered saline (PBS; pH 7.4) for 30 min at room temperature to inactivate endogenous peroxidase. Sections were pretreated with 3% bovine serum albumin in PBS for 30 min at room temperature, followed by incubation with primary antibodies against Shh (1:100, Bioss, Woburn, MA, USA), Gli1 (1:100, Novus Biologicals, Centennial, CO, USA), Osx (1:1000, Abcam, Cambridge, MA, USA), Runx2 (1:1000, MBL, Nagoya, Japan), and OPN (1:100, antibody was kindly provided by Dr. Hiroaki Nakamura, Matsumoto Dental University, Japan) overnight at 4 °C. Sections were reacted with Histofine Simple Stain rat MAX-PO (MULTI; Nichirei, Tokyo, Japan) for 1 h at room temperature. Color was developed using liquid diaminobenzidine substrate-chromogen system (Dako, Carpinteria, CA, USA). Immunostained sections were then counterstained with methylene green.

### 3.6. Image Analysis

The 500 μm portion from the fracture line of the cortical bone to the midshaft was defined as the periosteum near the fracture site, and the 2000 μm portion from the fracture line of the cortical bone to the midshaft was defined as the periosteum far from the fracture site. The number of positive cells of Runx2, Osx, Shh, and Gli1 present in the periosteum near and far from the fracture were counted by defining a square (100 × 100 μm^2^).

### 3.7. Statistical Analysis

The statistical analysis of the data gathered from Runx2-, Osx-, Shh-, and Gli1-positive cell counting was performed using SPSS version 23 (SPSS Inc., Chicago, IL, USA). Analyses of variance were followed by the *t*-test and Tukey’s test to determine significance.

## 4. Conclusions

To date, it has been demonstrated that Ihh participates in fracture healing by promoting chondrocyte differentiation. Ihh signaling critically regulates osteoblast differentiation during endochondral bone development after bone fracture [19,23]. Our results demonstrate that the Shh–Gli1 signaling pathway is involved in intramembranous and endochondral ossification during the fracture healing process.

## Figures and Tables

**Figure 1 ijms-21-00677-f001:**
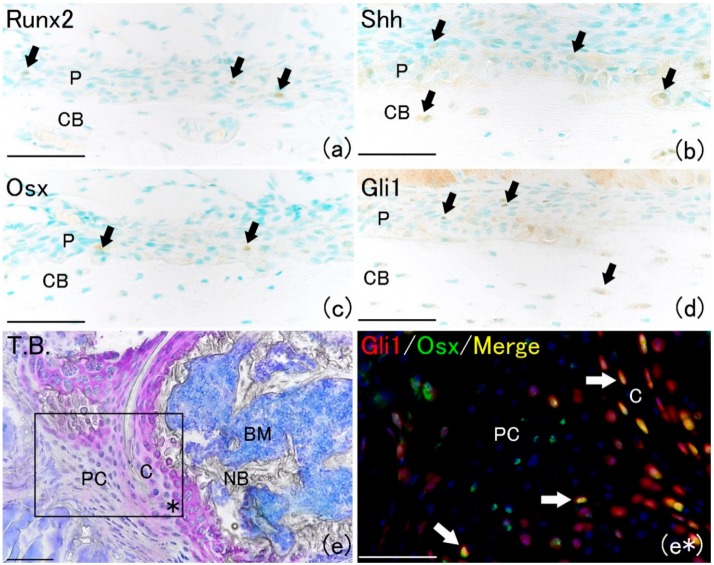
Histological analysis at day 0 on rat lib bone and at day seven on mouse lib bone fracture. (**a**) Runx2-positive cells were rarely observed at the surface of bone matrix in the periosteum (arrows). Scale bar: 50 μm. (**b**) Shh-positive expression was observed at the surface of bone matrix and osteocyte (arrows). Scale bar: 50 μm. (**c**) Osx-positive cells were rarely observed at the surface of bone matrix in the periosteum and same localization as Runx2. Scale bar: 50 μm. (**d**) Gli1-positive expression was observed at the surface of bone matrix and osteocyte and same as Shh localization (arrows). Scale bar: 50 μm. (**e**) Cross section pictures of fracture site. Newly formed cartilage matrix was observed around new bone and bone marrow. Scale bar: 100 μm. (**e***) Gli1-positive cells were observed in perichondrium and formed new cartilage at the fracture site (red fluorescent cell). Osx-positive cells were localized around the new cartilage matrix (green fluorescent cell). The merged image of Gli1- and Osx-positive areas demonstrated that the most Osx-positive cells merged on Gli1-positive cells (arrows). Scale bar: 50 μm. (P: periosteum; CB: cortical bone; PC: perichondrium; C: cartilage; NB: new bone; BM: bone marrow).

**Figure 2 ijms-21-00677-f002:**
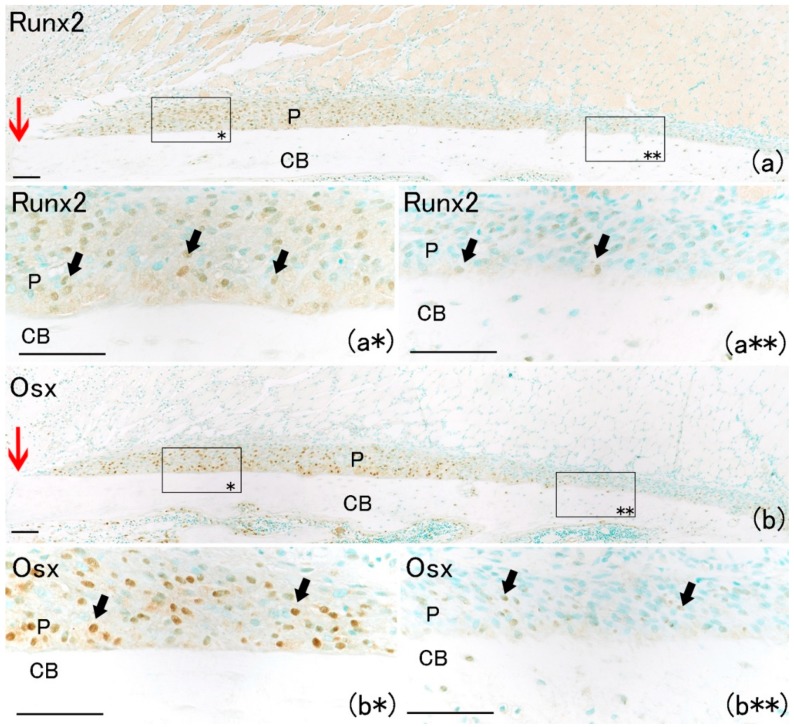
Histological analysis at day 1 on rat lib bone fracture (red arrow). (**a**) Runx2-positive cells were observed in remained periosteum. Scale bar: 100 μm. (**a***) Runx2-positive cells near the fracture site (arrows). Scale bar: 50 μm. (**a****) Runx2-positive cells were rarely observed far from bone fracture site. Scale bar: 50 μm. (**b**) Osx-positive cells were observed in remaining periosteum. Scale bar: 100 μm. (**b***) Osx-positive cells near the fracture site (arrows). Scale bar: 50 μm. (**b****) Osx-positive cells were rarely observed far from the bone fracture site. Scale bar: 50 μm. (P: periosteum; CB: cortical bone).

**Figure 3 ijms-21-00677-f003:**
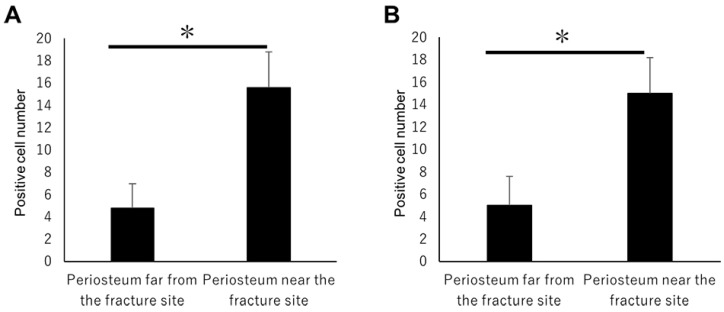
The positive cell number count on day 1. (**A**) Runx2-positive and (**B**) Osx-positive cell numbers in the periosteum near the fracture site were remarkably higher than those far from the fracture site. Data represent the mean ± SE (n = 3/group). Asterisks indicate the statistical significance of the differences (* *p* < 0.05, *t*-test).

**Figure 4 ijms-21-00677-f004:**
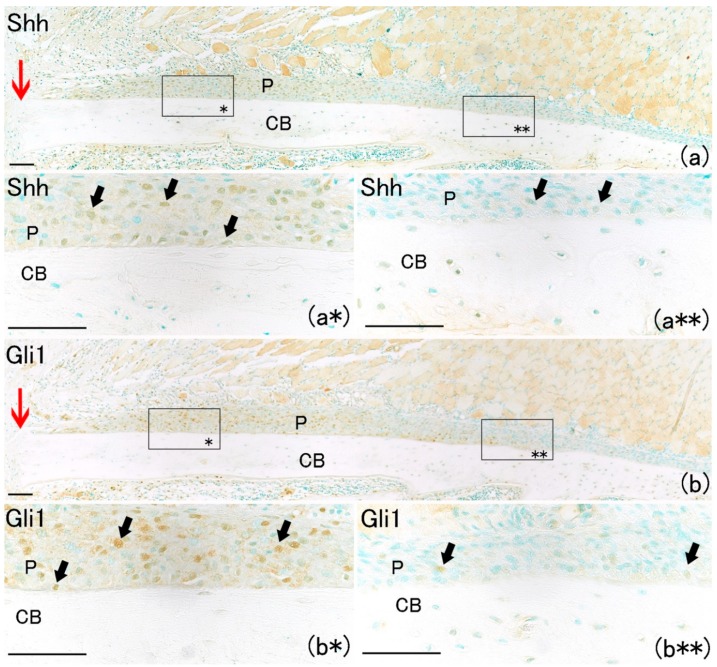
Histological analysis at day 1 on rat lib bone fracture (red arrow). (**a**) Shh-positive cells were observed in remaining periosteum. Scale bar: 100 μm. (**a***) Shh-positive expression near the fracture site (arrows). Scale bar: 50 μm. (**a****) Shh-positive cells were rarely observed far from the bone fracture site. Scale bar: 50 μm. (**b**) Gli1-positive cells were observed in remaining periosteum. Scale bar: 100 μm. (**b***) Gli1-positive cells near the fracture site (arrows). Scale bar: 50 μm. (**b****) Gli1-positive cells were rarely observed far from the bone fracture site. Scale bar: 50 μm.

**Figure 5 ijms-21-00677-f005:**
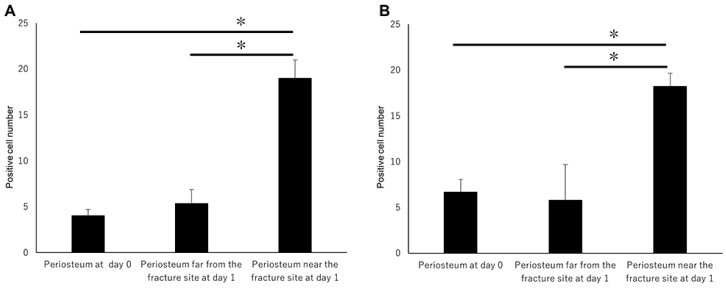
The positive cell number count on day 1. (**A**) Shh-positive and (**B**) Gli1-positive cell numbers in the periosteum near the fracture site were remarkably higher than those far from the fracture site. (A) Shh-positive and (B) Gli1-positive cell numbers in the periosteum near the fracture site at day 1 were higher than those at day 0. Data represent the mean ± SE (*n* = 3/group). Asterisks indicate the statistical significance of the differences (**p* < 0.01, Tukey’s test).

**Figure 6 ijms-21-00677-f006:**
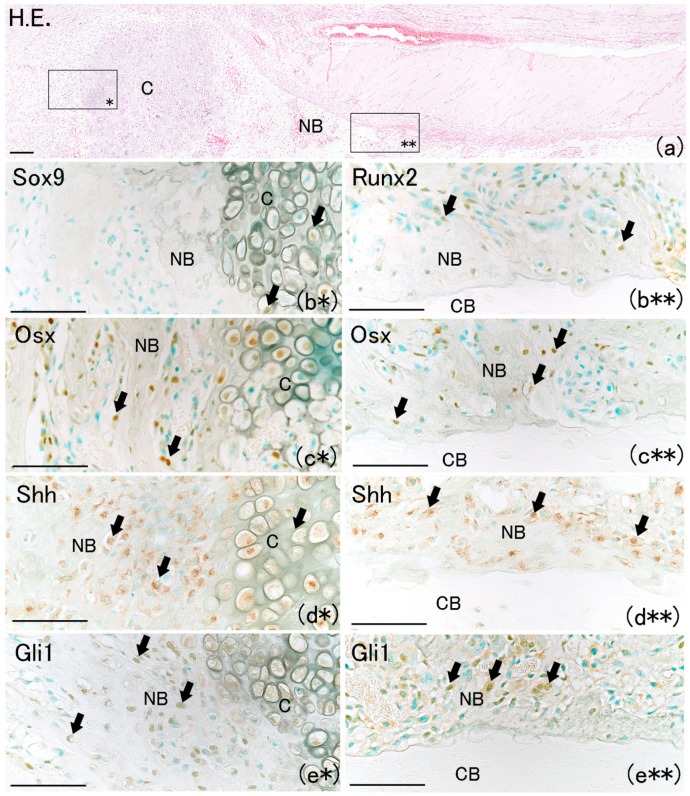
Newly formed cartilage and bone matrix were observed in the fracture site. (**a**) Newly formed cartilage matrix was observed in fracture site at day 7 on rat lib bone fracture. Scale bar: 100 μm. (**b***) Sox9-positive cells were observed in chondrocyte but not on the cell surface of bone matrix. Scale bar: 50 μm. (**c***) Osx-positive cells were observed at the surface of newly formed bone matrix and in chondrocyte (arrows). Scale bar: 50 μm. (**d***) Shh-positive expression was also around the new cartilage matrix (arrows). Scale bar: 50 μm. (**e***) Gli1-positive cells were also around the new cartilage matrix (arrows). Scale bar: 50 μm. (**b****) Runx2-positive cells at the surface of new bone but in the existing bone (arrows). Scale bar: 50 μm. (**c****) Osx-positive cells at the surface of new bone but in the existing bone (arrows). Scale bar: 50 μm; (**d****) Shh-positive expression at the surface of new bone but in the existing bone (arrows). Scale bar: 50 μm. (**e****) Gli1-positive cells at the surface of new bone but in the existing bone (arrows). Scale bar: 50 μm. (CB: cortical bone; NB: newly formed bone; C: cartilage).

**Figure 7 ijms-21-00677-f007:**
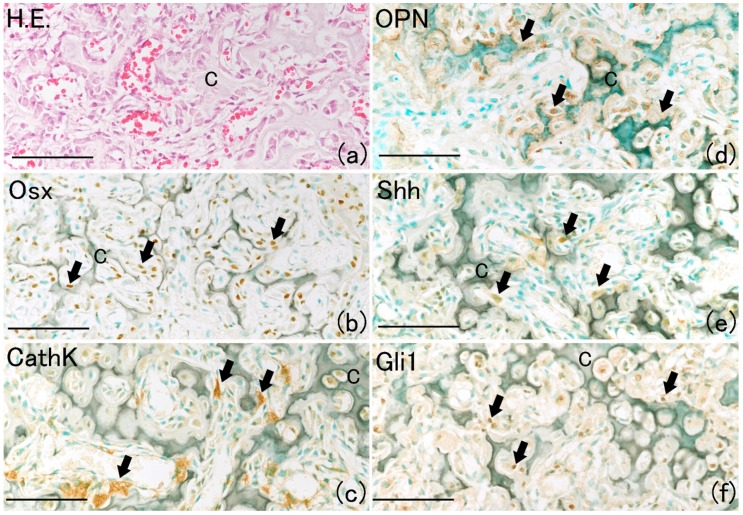
Histological analysis at day 14 on rat lib bone fracture. (**a**) Reduction of formed cartilage matrix began to resorb and be replaced by new bone formation on rat lib bone fracture. (**b**) Osx-positive cells were observed around reduced cartilage matrix (arrows). (**c**) Localization of CathK-positive cells was around remaining cartilage matrix. (**d**) OPN-positive area was observed around the cartilage matrix. (**e**) Shh-positive areas localized around cartilage matrix (arrows). (**f**) Gli1-positive areas found around cartilage matrix (arrows). (NB: newly formed bone; C: cartilage; Scale bar: 50 μm).

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
