# Peer review of "Sonic Hedgehog Regulates Bone Fracture Healing"

_ijms, 2020, doi:10.3390/ijms21020677_

Round 1
Reviewer 1 Report
Bone fracture healing involves the combination of intramembranous and endochondral ossification. It is known that Ihh promotes chondrogenesis during fracture healing. Meanwhile, Shh has been reported to be involved in fracture healing, but the details were not clarified. The authors demonstrated that Shh participates in fracture healing. The localization of Shh and Gli1 during fracture healing was examined. The localization of Gli1 progeny cells and osterix (Osx)-positive cells was similar during fracture healing. Shh and Gli1 were co-localized with Runx2 and Osx. These findings suggest that Shh is involved in intramembranous and endochondral ossification during fracture healing.
The manuscript is interesting and concisely written, but there are several critical issues.
How many rats and mice used? Why were only males of rats and mice used? Six-week-old male Sprague-Dawley rats (Hokudo, Sapporo, Japan) and Gli-CreERT2; tdTomato male mice (Jackson Laboratory, Bar Harbor, ME, USA) were used.Immunohistochemistry; it is not quantitative method, It should be better to have quantitative method with statistical analysis.
Analyze the expression levels of Ihh, Shh and Gli1 by qPCR or ther alternative methods. Any reason why both mice and rats were used? Only mice may not be enough. Specify the reason. Specify the pathway in details on how Ihh and others are associated (including figure). There are many abbreviations; add the list.Overall the manuscript should be extensively revised.
Author Response
Reply to Reviewer 1:
Dear Sir:
We greatly appreciate your critical reading and constructive comments on our manuscript. We have revised the manuscript on the basis of your suggestions. As indicated in the responses that follow, we have taken all these comments and suggestions into account in the revised version of our paper.
Comment #1
How many rats and mice used? Why were only males of rats and mice used? Six-week-old male Sprague-Dawley rats (Hokudo, Sapporo, Japan) and Gli-CreERT2; tdTomato male mice (Jackson Laboratory, Bar Harbor, ME, USA) were used. Immunohistochemistry; it is not quantitative method, It should be better to have quantitative method with statistical analysis.
Response
As you have mentioned, we have added the number of rats and mice (page 9, line 206). To avoid errors in the experimental data owing to the estrous cycle, all the animals used in this study were male. As recommended, we statistically examined the difference in the number of Runx2, Osterix, Shh, and Gli1-positive cells in the periosteum near and far from the fracture site. We have added the method of counting the number of positive cells and the method of statistical analysis in the “Materials and Methods” section (page10, lines 241–250). The results are shown in Figures 3 (page 5, lines 132–136) and 5 (pages 6–7, lines 152–157). In addition, we have added and corrected the text regarding Figure 3 (page 4, lines 120–121) and Figure 5 (page 5, lines 139–143).
Comment #2
Analyze the expression levels of Ihh, Shh and Gli1 by qPCR or ther alternative methods. Any reason why both mice and rats were used? Only mice may not be enough. Specify the reason.Specify the pathway in details on how Ihh and others are associated (including figure). There are many abbreviations; add the list. Overall the manuscript should be extensively revised.
Response
In this study, we have tried immunostaining for Ihh, but we did not obtain a good staining result. Nonetheless, as you have said, it is important to do a quantitative search on how Ihh is related to Shh and Gli1. Therefore, as an alternative method, we counted the number of Shh- and Gli1-positive cells on days 0 and 1 and statistically examined the difference in the number of the positive cells. We have also added the method of counting the number of positive cells and the method of the statistical analysis in the “Materials and Methods” section (page 10, lines 241–250). The results are shown in Figure 5 (pages 6–7, lines 152–157). In addition, we have added and corrected the text regarding Figure 5 (page 5, lines 139–143). As recommended, we have also added the abbreviations list (page 11, line 255). We have used a rat fracture model in this animal experiment. Nonetheless, it was necessary to examine whether the progeny of Gli1-positive cells was related with osteoblast differentiation. Therefore, Gli-CreERT2; tdTomato mice were used in this study.
We hope that these modifications meet your requirements, and we thank you again for your helpful comments.
Sincerely yours,
Hiroaki Takebe, DDS., Ph.D.
Department of Histology, Health Sciences University of Hokkaido, Ishikari-Tobetsu, Hokkaido 061-0293, Japan
E-mail: takebeh@hoku-iryo-u.ac.jp
Reviewer 2 Report
Page 2 line 51: At the end of the sentence “These pathways are also activated during intramembranous and endochondral ossification in the fracture healing process” consider adding the phrase “but it is not clear if they are involved in the heling process”.
Page 2 Line 70: This paper would be clearer if you gave a hypothesis and the state that you would use other markers such Osx and Runx2 to follow those relationships. Then, insert lines 72 to 74 in this location.
Page 3 Line 81-82: I Think that the sentence “In our genetically modified mouse system, both Gli1-positive cells and their progeny were permanently marked by red fluorescent protein expression.” Shows that you were able to track the cells. I am not certain if you could track them prior to cre expression 3 days after tamoxifen and identify the same cells day 7. For those of us not using cre mice a brief explanation would help. I suspect that the cre helped you identify progeny cells.
Page 3 Lines 106-111: The merging of these results with discussion is wonderful. It puts the significance in immediate context.
Author Response
Reply to Reviewer 2:
Dear Sir:
We first express our thanks for your helpful suggestions.
In response to your comments, we have made the following revisions.
Comment #1
Page 2 line 51: At the end of the sentence “These pathways are also activated during intramembranous and endochondral ossification in the fracture healing process” consider adding the phrase “but it is not clear if they are involved in the heling process”.
Response
As you have recommended, we have added the phrase “but it is not clear if they are involved in the healing process” at the end of the sentence “These pathways are also activated during intramembranous and endochondral ossification in the fracture healing process” (page 2, line 51).
Comment #2
Page 2 Line 70: This paper would be clearer if you gave a hypothesis and the state that you would use other markers such Osx and Runx2 to follow those relationships. Then, insert lines 72 to 74 in this location.
Response
As you have mentioned, we have added the hypothesis and the statement regarding this study (page 2, lines 72–74).
Comment #3
Page 3 Line 81-82: I Think that the sentence “In our genetically modified mouse system, both Gli1-positive cells and their progeny were permanently marked by red fluorescent protein expression.” Shows that you were able to track the cells. I am not certain if you could track them prior to cre expression 3 days after tamoxifen and identify the same cells day 7. For those of us not using cre mice a brief explanation would help. I suspect that the cre helped you identify progeny cells.
Response
As you have mentioned, in this study, we were able to track the Gli1-positive cells and their progeny with the use of the genetically modified mice, and we added a description regarding this principle (page 2, lines 86–93). As you have said, it is impossible to track the Gli1-positive cells before tamoxifen administration. Moreover, we have confirmed that red fluorescence was appropriately expressed when tamoxifen was administered to the genetically modified mice for three consecutive days by preliminary experiments.
Comment #4
Page 3 Lines 106-111: The merging of these results with discussion is wonderful. It puts the significance in immediate context.
Response
As you have recommended, we statistically examined the difference in the number of Runx2- and Osterix-positive cells in the periosteum near and far from the fracture site. The results are shown in Figure 3 (page 5, lines 132–136).
We hope that the above modifications meet your requirements and thank you again for your helpful comments.
Sincerely yours,
Hiroaki Takebe, DDS., Ph.D.
Department of Histology, Health Sciences University of Hokkaido, Ishikari-Tobetsu, Hokkaido 061-0293, Japan
E-mail: takebeh@hoku-iryo-u.ac.jp
Round 2
Reviewer 1 Report
The authors responded to the reviewers' comments in a proper way.